# Monitoring of Single-Track Melting States Based on Photodiode Signal during Laser Powder Bed Fusion

**DOI:** 10.3390/s23249793

**Published:** 2023-12-13

**Authors:** Longchao Cao, Wenxing Hu, Taotao Zhou, Lianqing Yu, Xufeng Huang

**Affiliations:** 1School of Mechanical Engineering & Automation, Wuhan Textile University, Wuhan 430200, China; clc@wtu.edu.cn (L.C.); wenxinghu37@gmail.com (W.H.); 2Hubei Key Laboratory of Digital Textile Equipment, Wuhan Textile University, Wuhan 430200, China; 3China Ship Development and Design Center, Wuhan 430200, China; kttktt@163.com; 4School of Aerospace Engineering, Huazhong University of Science & Technology, Wuhan 430074, China; huangxufeng@hust.edu.cn

**Keywords:** deep learning, in situ monitoring, laser powder bed fusion, photodiode signal, convolutional neural network

## Abstract

Single track is the basis for the melt pool modeling and physics work in laser powder bed fusion (LPBF). The melting state of a single track is closely related to defects such as porosity, lack of fusion, and balling, which have a significant impact on the mechanical properties of an LPBF-created part. To ensure the reliability of part quality and repeatability, process monitoring and feedback control are emerging to improve the melting states, which is becoming a hot topic in both the industrial and academic communities. In this research, a simple and low-cost off-axial photodiode signal monitoring system was established to monitor the melting pools of single tracks. Nine groups of single-track experiments with different process parameter combinations were carried out four times and then thirty-six LPBF tracks were obtained. The melting states were classified into three classes according to the morphologies of the tracks. A convolutional neural network (CNN) model was developed to extract the characteristics and identify the melting states. The raw one-dimensional photodiode signal data were converted into two-dimensional grayscale images. The average identification accuracy reached 95.81% and the computation time was 15 ms for each sample, which was promising for engineering applications. Compared with some classic deep learning models, the proposed CNN could distinguish the melting states with higher classification accuracy and efficiency. This work contributes to real-time multiple-sensor monitoring and feedback control.

## 1. Introduction

Additive manufacturing (AM)—also known as 3D printing—has experienced exponential development in recent years, which is becoming increasingly attractive to many industrial fields, especially the aerospace and biomedical industries [1,2,3,4]. Laser powder bed fusion (LPBF) is treated as one of the most hopeful metal-based AM technologies for its prominent performance in fabricating parts with complex and fine inner structures [5]. Nevertheless, an LPBF-created part can suffer several different types of defects (porosity, poor surface finish, cracking, etc.) due to processing parameter fluctuations or unstable processing conditions, which hinder its widespread application [6,7]. In recent years, in situ monitoring has emerged as an alternative accommodation for defect identification and control, which has become a research hot topic [8].

One main reason that causes the above-mentioned defects is improper energy input during the LPBF [9]. For example, it is easier to cause balling when the inputted energy density is insufficient. Higher energy density can cause spatter and overheating of the metal powder, which has a bad impact on the mechanical properties of the LPBF parts [10]. To predict the quality of an as-built part, it is crucial to distinguish various process states and monitor the defects during the LPBF [11,12,13]. Numerous studies have been conducted to monitor states and detect defects based on different signals [14,15,16]. Ye et al. [17] proposed a state monitoring method using a deep belief network (DBN) based on the acoustic signal output during the LPBF process. The DBN model could achieve an average accuracy of 93% in recognizing five melting states of single tracks. Zhang et al. [18] used a high-speed camera to acquire process images and inputted the extracted feature vectors into the convolutional neural network (CNN) model and Support Vector Machine (SVM) to classify the three processing states. The accuracy of the CNN model could reach 92.8%. Zhang et al. [19] decomposed the original signal from the light signals using the wavelet packet decomposition (WPD) method. Based on these real-time features, a DBN method was established to identify four welding states. The average accuracy of defect detection was above 90%. Pandiyan et al. [20] built an explainable machine-learning model based on the acoustic emission signal to identify the three regimes: lack of fusion mode, conduction mode, and keyhole mode. It implies that process monitoring is a feasible measure for state identification, while the accuracy needs to be improved.

The process of laser interacting with metal powder is a complex thermal process along with various physical and chemical changes. The optical, acoustic, and thermal signals are generated from the molten pool, spatter, and metal vapor [2,21]. These signals contain abundant closely related information on defect formation [22,23,24]; therefore, the light signal, monitored using photodiode sensors, attracts attention. Bono et al. [25] used four photodiode sensors at different wavebands to collect visible and infrared light signals through different bandpass filters. The correlations between photodiode signal and porosity defects were established. Coeck et al. [26] proposed pseudo-color maps using the photodiode signals to predict the size and position of pores. The prediction accuracy was 90% for pores with a volume greater than 0.001 mm^3^. Okaro et al. [27] detected the light signal using a photodiode sensor while building 49 tensile test bars. A semi-supervised learning method was utilized to categorize each bar quality according to the ultimate tensile strength. Montazeri et al. [28] identified the material cross-contamination caused by the introduction of trace foreign matter into the raw powder materials using the photodiode sensor during the LPBF. Zhang et al. [29] studied the influence of laser power on the molten pool behavior of ceramics in a single track of the LPBF. The correlations between the photodiode signal and the delay of the scanner, the ‘edge effect’, and the unstable temperature field were analyzed. Nadipalli et al. [30] converted the time series photodiode signals into spatial coordinates. The robustness and sensitivity between the co-axial and off-axial setup for defect identification were contrasted. Ding et al. [31] used a photodiode to inspect the dynamic behavior of the molten pool and plume. Lapointe et al. [32] built a data-driven model using experimental diagnostics for the optimization of laser process parameters prior to printing, which led to improved part quality. The above studies demonstrate that the photodiode is an effective and superior method for monitoring the LPBF process.

However, the fast scanning speed of LPBF puts forward strict requirements on the data sampling rate and processing efficiency. Traditional data analysis methods mainly including time-frequency transformation and manual feature extraction are not adequate [33]. So far, the rapidly developed deep learning method has been extensively applied to monitor the LPBF process [34,35]. Meanwhile, compared to the expensive high-speed camera and infrared images, the photodiode can cover the wavelength range of the light emitted from the molten pool, spatter, and plume. Considering the significant impact of the melting state of the powder bed on the mechanical properties, it is desirable to monitor the melting state by combining the photodiode signals and a deep learning method. In this work, an off-axial photodiode monitoring system was established to monitor the LPBF process. A total of 36 LPBF tracks and their corresponding photodiode signals were obtained. The signals were labeled into three classes according to the track morphologies. A CNN model was developed to identify the melting states based on the collected photodiode data.

This paper is organized as follows: Section 2 introduces the experimental setup and the photodiode signal monitoring platform. The acquired data during LPBF were preliminary analyzed. In Section 3, the acquisition and processing of datasets and the constructed CNN are introduced. In Section 4, the performance of the CNN model for LPBF melting states monitoring is analyzed and compared with other network models, followed by the conclusion and future work in Section 5.

## 2. Experiment Equipment and Datasets

### 2.1. LPBF Machine and Powder Materials

An LPBF machine of FastForm140 was utilized for the experiments. The maximum scanning speed can reach 7000 mm/s and the range of the coating layer thickness is between 10 and 40 µm, approximately. The LPBF equipment is mainly composed of six parts including a control system, a powder feeding system, a fiber laser, a preheating system, a galvanometer system, and a water cooling system. The whole process was carried out in a sealed building chamber. A total of 99.9% Ar was used as a shielding gas with a fixed flow rate of 1.0 L/min to keep the level of oxygen under 4000 ppm. The control panel was utilized to set the process parameters (the scanning speed, laser powder, hatch, oxygen content, etc.). The powder feeding system was used to supply, spread, and recycle the feedstock. The fiber laser, with a maximum power of 250 W, was applied as the heat source. The beam profile was Gaussian with a wavelength of 1080 nm. The powder and substrate were preheated to a set temperature (about 473.15 K) using the preheating system. The partial basic configuration parameters of the LPBF machine are listed in Table 1.

The substrate was made of 316 L stainless steel with a thickness of 20 mm. The 316 L stainless steel powder was chosen for the advantages of relatively low cost and good performance. The research on this material is relatively mature and it has been widely adopted in the biomedical, nuclear, aerospace, and marine industries [36]. The powders were spherical and the diameter of the powders ranged from 15 to 37 μm. The main chemical compositions of the feedstock are listed in Table 2.

### 2.2. Photodiode-Based Signals Acquisition System

A photodiode sensor-based monitoring system was used for the following reasons: (a) The photodiode sensor can reach a fast data collection rate at a relatively low cost; (b) The size of the photodiode sensor is small, which is convenient to install in or out of the building chamber while the camera is always installed out of the chamber; (c) The post-processing requirements of the camera-collected images make it difficult to apply these developmental monitoring methods to production systems in real-time; (d) The replication of results from camera-based monitoring systems in a production environment is limited by the need for specialized hardware with high computational capacity [37]; (e) The continuous monitoring time is limited for some high-speed cameras. It is hard to reach continuous real-time monitoring during the building of a complex part. Furthermore, the information from a single sensor is limited, comprehensive information needs to be collected using multiple sensors. The photodiode can be an ideal sensor for multi-sensor monitoring. It needs the signal-to-image conversion for fusion of the information obtained using the camera. As illustrated in Figure 1a, the photodiode sensor-based monitoring system includes the following components: a photodiode sensor, a DC power supply, a data acquisition card (DAQ), a personal computer, and the corresponding software system. The photodiode sensor is produced by Hamamatsu with a model of C10439-11 (shown in Figure 1c). The response wavelength range was 500 nm to 1700 nm and the peak sensitivity wavelength was 1550 nm. The adopted data acquisition card used was NI 9221 with a maximum sampling rate of 80 kHz. The data acquisition card was connected to the personal computer. The NI-DAQmx driver and Labview software 2019 were applied to set the acquisition parameters.

### 2.3. Data Collection and Track Analysis

As shown in Figure 1b, the photodiode sensor was placed inside the build chamber at a distance of about 300 mm from the molten path. It was fixed on a magnetic holder installed beside the overflow container. Nine pairs of processing parameters were intentionally selected to produce scan tracks with three melting states after trial and error. The sampling rate of the photodiode was set to 50 kHz/s to collect detailed characteristic information during the single-track experiments. A track with the same combination of processing parameters was printed four times to reduce the uncertainty because of fluctuations and acquire sufficient data to train the deep learning models. Therefore, a total of 36 single-track experiments were performed. All the tracks were 60 mm in length. The spacing distance between the center of adjacent tracks was 2 mm. The layer thickness was fixed at 50 μm.

The melting states of single tracks are determined by the volumetric energy density E [38,39], which is defined as
(1)$$E=\frac{P}{\pi r^{2}v}$$
where P denotes the laser power; r presents the radius of the focal spot on the surface of the powder bed; and v denotes the scanning speed. The laser energy alone has a decisive impact on the morphology of a track. Optical microscopy and a laser confocal microscope were used to observe the morphology of the tracks. As shown in Figure 2, the three typical melting states—lack of melting (LOM), normal melting (NM), and over-melting (OM)—were defined according to the morphology of the scan tracks. The processing parameters and the melting states are shown in Table 3.

The acquired signals from the photodiode were labeled into three categories to represent the melting states. As shown in Figure 2a and Figure 3a, the lack of a melting track is narrow and discontinuous while the edge of the track is irregular because the volumetric energy density is not enough to fuse the powder layer. As demonstrated in Figure 2c and Figure 3e, the over-melting track is deeper and wider because the volumetric energy density is higher. A recoil pressure from the keyhole is higher to remove adjacent powder and molten metal resulting in the penetration of the substrate. The normal track is smoother and the edge is relatively regular as shown in Figure 2b and Figure 3c. The acquired partial-light data of corresponding melting states are demonstrated in Figure 3b,d,f. The higher the laser power was, the bigger the fluctuation range in the photodiode signal values, and the less stable the molten pool became. The laser power increase means an increase in the input energy. As the input energy increases, the fluctuation of the photodiode signal becomes bigger. This result also shows good agreement with that of Zhang [29]. To distinguish the melting states just using the signals manually is not practical. A deep learning-based method of feature extracting and melting-state identification is desired.

## 3. Data Processing and CNN Model

### 3.1. Data Processing

A CNN structure is constructed to identify the states of the scan track based on a photodiode sensor. The row photodiode signal is inputted to the CNN model without signal pre-processing. As shown in Figure 4, the developed approach mainly consists of three steps: data collection, signal-to-image transformation, and CNN-based melting state identification.

In step 1, an in situ monitoring device based on a photodiode sensor is established in Section 2.2. Then, thirty-six groups of experiments with nine combinations of processing parameters were conducted to produce melting tracks with three types of morphology states. During the LPBF process, the emitted light signal was obtained and saved. The melting state was identified according to the morphology of the tracks. Then, the acquired photodiode data were divided and labeled into three groups. The saved photodiode signal during the LPBF processing was sampled for the next step. In step 2, the raw one-dimensional light data were transformed into two-dimensional grayscale images. In step 3, alternating convolution and pooling operations were performed to automatically extract high-level features from transformed 2D grayscale images. Then, the extracted features from the photodiode signal were utilized to train a 2D CNN model to realize state identification.

### 3.2. Signal-to-Image Transformation Methodology

In most in situ monitoring methods, the original data would be filtered or smoothed before inputting into a deep learning model. The noise was reduced to some degree while some useful information could be also removed. In this work, a method called signal-to-image proposed by Li et al. [40] was applied to transform the raw 1D time-domain photodiode signal into 2D grayscale images. This signal-to-image method can effectively reduce the complicated and time-consuming handcraft feature extraction. At the same time, the converted grayscale images inputted into the CNN model are beneficial to take advantage of the CNN model. The procedures of signal-to-image transformation are shown in step 2 of Figure 4.

A running window with a length of $M^{2}$ was taken to slide on the raw 1D photodiode data with a sliding step L. After that, the original data were split into samples with a size of $M^{2}$. The image with pixels of $M^{2}$ was filled with the time-domain photodiode data point sequentially. Then, the image data sample was normalized to a range of 0 to 255. The process of normalization can be expressed by Equation (3). The R(j,k) denotes the pixel value in the *j*th row and *k*th column of an image; round(·) is the rounding function; $L\left(i\right),i=1,2,\ldots ,M^{2}$ is the *i*th sample point in the raw photodiode signal; and max(L) and min(L) are the maximum and minimum values in the original data, respectively. It should be noted that the value of *M* has an important influence on the computational cost. In this work, the values of *M* are set as 32 in terms of the computational time.
(2)$$R(j,k)=round\left\{\frac{L((j-1)\times M+k)-\min(L)}{\max(L)-\min(L)}\times 255\right\},(j,k=1,2,\ldots ,M)$$

According to Equation (2), the raw photodiode data were transformed into 2D grayscale images. The size of the running window was 32 × 32 and the sliding step was 170 data points. To make the data of different categories balance, 34,854 data points of each track were used to convert into 2D grayscale images. In the end, 1,254,744 data points were applied to generate 7200 grayscale images.

The images denoting the three melting states are demonstrated in Figure 5. The first row presents the track morphologies measured using laser confocal microscopy, the second row presents the raw data, and the last row demonstrates the corresponding transformed images. The three columns demonstrate three categories of LOM, NM, and OM. It is found that the gray value and gray level distribution have outstanding differences for the three kinds of melting states.

All the collected signals from experiments were scanned running windows, of which the 1D data were transformed into 2D grayscale images that were used as inputs of the CNN classifier. To make the data of different categories balance, 34,854 successive data points of each track experiment were applied to convert into 2D grayscale images, and then a total of 1,254,744 data points were augmented from 36 groups of experiments. Therefore, 200 images were obtained for each track and a total of 7200 images were obtained for three categories states. A total of 80% of the converted images in each class were stochastically selected for training (5760) and the remaining 20% for testing (1440). In this way, there were sufficient training samples to build and validate the state classification models.

### 3.3. General CNN Model

The recognition and classification of the melting states were originally a non-linear dimensionality problem. The General CNN model consists of a typical architecture including of input layer, convolutional (Conv) layer, pooling layer, fully connected layer, and output layer. The Conv layer performs a convolution operation to extract the features from the input tensor. The convolution operation can be expressed by the following formula [41,42,43]:(3)yconv=f((W∗X)+b)
where yconv denotes the output tensor of the Conv layer; X denotes the input tensor; W presents the convolution kernel consisting of weights; ‘∗‘ is the convolution operator; *b* denotes an additive bias; and f() is an activation function. The sigmoid function, Tanh function, and rectified linear unit (ReLU) function are the most popular activation functions; the ReLU was adopted in this proposed CNN structure.

To control over-fitting—by progressively decreasing the spatial size of the representation and the number of parameters while saving the computation time in the CNN structure—a pooling operation was performed to summarize the features of a region with a value. Common pooling methods include average pooling and maximum pooling. In this work, the maximum pooling method was adopted. It can be expressed as
(4)$$ypool=\max(w(s_{1},s_{2})\cap yconv)$$
where ypool denotes the output tensor of the pooling layer; $s_{1}$ and $s_{2}$ denotes the length and the width of the pooling window size, respectively; and $w(s_{1},s_{2})$ presents the pooling window, which can be set to 3 × 3, 5 × 5, or 7 × 7.

The CNN model consisted of several Conv and pooling layers to accomplish a high accuracy. The output tensor after several convolutional and pooling operations is inputted into the fully connected layer. The fully connected layer is also called the classification layer which acts as a classifier. The fully connected layer connects each neuron in one layer to the next layer, which is expressed as
(5)$$y=f(W_{f}y^{f}+b_{f})$$
where y presents the output matrix; $W_{f}$ denotes the weight; $y^{f}$ denotes the output of the previous layer; and $b_{f}$ is the bias of the fully connected layer neurons.

The output layer is behind the fully connected layer, which is performed to transform the last layer into the probability of each class. It can be expressed as
(6)$$Y=\sigma (y^{L})$$
where Y denotes the matrix of the predictive results; $y^{L}$ presents the results of the final fully connected layer; and σ(·) is the logistic function to normalize $y^{L}$ into a probability distribution. The softmax was applied for multi-class classification.

### 3.4. Construction of the Proposed CNN

A two-dimensional convolutional neural network is constructed. The framework of the developed 2D CNN model is demonstrated in Figure 6. The number of convolution layers, pooling layers, and fully connected layers for all is three. The first convolution layer is behind the input layer (a tensor of size 32 × 32 × 1) and each convolution layer is followed by a pooling layer. The three fully connected layers are successive behind the last pooling layer. The kernel size of the convolution is 3 × 3 with a stride of 1. The number of channels of the three convolution layers is 16, 64, and 128, respectively. The max-pooling is adopted in all the pooling layers with a pool size of 2 × 2 and a stride of 2. The neurons of the three fully connected layers are 64, 512, and 3, respectively. A dropout of 20% was used followed by the first and second fully connected layers to avoid overfitting. A zero-padding operation was applied to keep the dimension. The Rectified Linear Unit (ReLU) was adopted as the activation function. The parameter configuration of the proposed CNN structure is shown in Table 4. Conv presents the convolution layer and FC denotes the fully connected layer.

The raw one-dimensional photodiode signals were transformed into grayscale images and inputted into the CNN model, as shown in Figure 4 and Figure 6. Each sliding window includes 1024 (32 × 32) data points in the time domain of the one-dimensional light signal, while a stride is the length of 170 data points. Then, the transformed grayscale images are inputted into the model. The characteristics are obtained in the convolution layers. After every convolutional operation, the characteristics are inputted into the pooling layer to decrease dimensionality. The convolution and pooling operations are applied alternately to extract the features deeply. After the convolution and pooling layers, there is a flattening operation, which is followed by the fully connected layers where all the feature maps are inputted. Ultimately, the classification results of the melting states of tracks are given by the output layer. Adam optimizer is adopted in the whole process. The self-adapting learning rate with a range from 3 × 10^−3^ to 3 × 10^−6^ is set with an initial value of 3 × 10^−3^ to search for the global optimal solution effectively. The number of epochs is set as 500 and the batch size is 64. An early stopping method is adopted to automatically stop training once the model performance stops improving on the validation dataset. The binary cross-entropy function is adopted as the loss function.

## 4. Results and Discussion

### 4.1. The Results of the Proposed CNN Model

The datasets were divided into three parts (‘lock of melting’, ‘normal melting’, and ‘over-melting’). A total of 80% of the data were randomly selected to train the built model structure, and the remaining 20% were applied to the test. Tenfold cross-validation was applied for error estimation. All the algorithms were performed in Python 3.8 with the TensorFlow library and run on Windows 10 with an Nvidia RTX 2080 GPU. The average classification accuracy was 95.81%, while the classification result is demonstrated as a confusion matrix. As shown in Figure 7, the horizontal axis has the predicted class while the vertical axis denotes the true class. The classification accuracy is defined as the number of true predictions divided by the total number of tests that lie along the diagonal of the matrix. Due to the excessive energy density and high laser power producing an extensive brilliant light signal—which may influence the data collection—the accuracy of the NM class is lower than the other classes. At the same time, the manual classification only through the top morphologies may not be very exact. About 7% of the normal melting tacks are misclassified as over-melting tracks, while about 5% of the over-melting tracks are mispredicted as normal melting tracks. This may be because the thickness of the single powder layer was not totally homogeneous in the practical experiments. The classification accuracy of lack of molting is close to 100%. All the same, the average accuracy of classification can reach 95.81% and the lowest accuracy is up to 91.98%, which exhibits great potential to be an effective way to monitor the LPBF process.

The evolutions of the accuracy and the loss during the training and testing are demonstrated in Figure 8. The training accuracy is close to 100% and the testing accuracy converges around 95%. The final testing accuracy is 95.81% at the epoch of 406. At the same time, the training and testing losses are close to zero, which implies the proposed CNN model is feasible. It should be noted that there is an irregularly violent fluctuation in the accuracy and loss curves. The reason is that the learning rate is self-adaptive in the range between 3 × 10^−3^ and 3 × 10^−6^. The fluctuation comes up once the learning rate is self-adjusted.

As shown in Figure 9, ten grayscale images are randomly selected to visualize the classification results intuitively. ‘true’ denotes the label, and ‘pred’ presents the predictive results. The number 0 presents the lack of melting state, 1 presents the normal melting state, and 2 denotes the over-melting state. It can be seen that no image is misidentified in the selected grayscale images, which exhibits the high accuracy of our constructed CNN model and its effectiveness.

To further estimate the effectiveness of the proposed CNN model, the t-distributed stochastic neighbor embedding (t-SNE) is adopted to obtain the two-dimensional representation of the classification. The t-SNE is a statistical method for visualizing high-dimensional data by giving each data point a location in a two- or three-dimensional map. In this work, the t-SNE is used to map the high-dimensional feature data point to a two-dimensional map. Figure 10 demonstrates the outputs of t-SNE computation with the input of the second fully connected layer data. It illustrates that the boundary is obvious among the three types of states, while a few overlaps exist between the ‘OM’ and ‘NM’ clusters. According to Equation (1), the NM state converts into the OM state as the volumetric energy density increases. Meanwhile, the excessive volumetric energy density and high laser power produce an extensive brilliant light signal, which makes it hard to identify the melting states when the volumetric energy density is at the junction. This reason also makes clear that there is no overlap at the boundary of LOM and NM. It indicates that the constructed model can effectively obtain and recognize the features of the photodiode signal of different melting states.

### 4.2. Comparison of Classic Deep Learning Models

To further evaluate the performance of the proposed model, the other classic models are performed based on the same datasets. The results of the classic models were compared with the proposed 2D CNN. Herein, a one-dimensional convolutional neural network (1D CNN) model, a recurrent neural network (RNN) model, and the classic VGG16 model were selected to compare with the proposed model in terms of accuracy, the computation time for the whole test set, and the efficiency. The efficiency was defined as the ratio of classification accuracy to computation time per sample. One-dimensional CNN is suitable for processing 1D signals [44]. RNN is very effective for sequential data [45]. It can extract temporal and semantic features from time-domain data. VGG16 is a deep convolutional neural network model proposed by K. Simonyan and A. Zisserman from the University of Oxford [46].

Table 5 lists the classification performance of each model. The performance of the compared models is demonstrated in Figure 11. For 1D CNN, the classification accuracy based on the raw data is 84.92% while the computation time of one sample is 81 ms. The RNN model has the lowest classification accuracy and efficiency, while its computation time is shorter than 1D CNN. VGG16 exhibits a better classification accuracy and efficiency coefficient, which are 87.30% and 2.36, respectively. The proposed CNN model has the highest classification accuracy of 95.81% and an efficiency of 6.38. Meanwhile, the computation time of the proposed CNN is the shortest. It indicates that the proposed CNN model exhibits excellent performance. However, it should be noted that if the time of signal-to-image transformation is included, the efficiency of the proposed CNN model will be reduced. The signal-to-image transformation is valuable when confused with the image signal in the scenario of multi-sensor monitoring.

## 5. Conclusions and Future Work

In this work, LPBF experiments were performed with different combinations of process parameters, and thirty-six tracks with three categories of melting states were obtained.

(1)An off-axis photodiode-based monitoring system was established to acquire the light signal while the tracks were melting. A method was used to convert the photodiode signal to grayscale images;(2)A CNN model was proposed to classify the melting state. Tenfold cross-validation was applied. The classification accuracy of the proposed CNN model can reach 95.81% with the shortest time of 15 ms for each sample;(3)The performance of the proposed model was compared to three classic deep learning methods (1D CNN, RNN, and VGG16). It demonstrates that the proposed model exhibits outstanding performance in terms of classification accuracy and efficiency;(4)It indicates that it is feasible and reliable to monitor the LPBF process using a simple and low-cost photodiode combined with the CNN model. This work can promote progress in monitoring the LPBF process and improving the reliability of component quality and the repeatability of manufacturing.

An as-built part consists of numerous tracks in various directions. The track melting states have an important influence on the properties of the as-built parts. However, only single tracks were analyzed and a single photodiode was applied. This work provides a feasible precept for monitoring the LPBF process to improve the quality of the LPBF-created part. At the same time, to improve accuracy and practicability, comprehensive information is needed and multi-sensor monitoring is a tendency. This work provides a manner of treating photodiode signals and fusing the multisource signals. In future work, the LPBF process of the whole complex part will be monitored with multiple sensors.

## Figures and Tables

**Figure 1 sensors-23-09793-f001:**
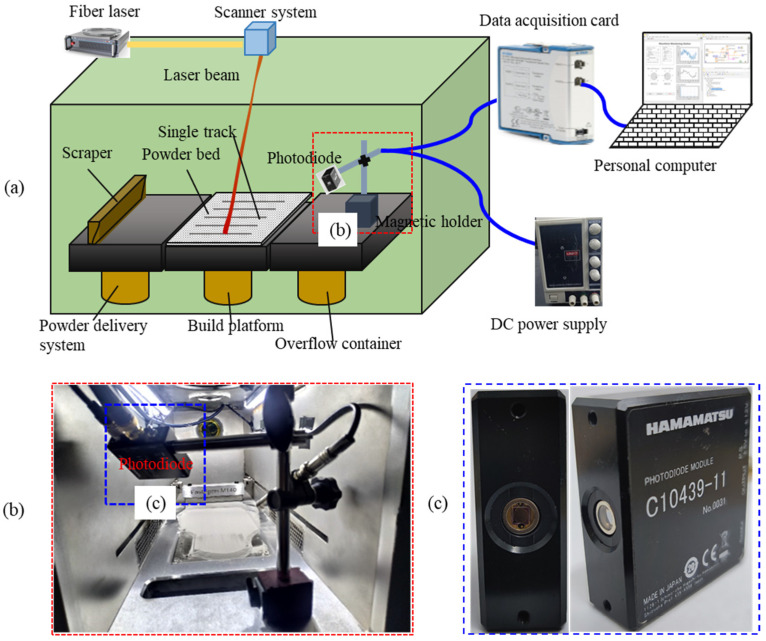
Photodiode sensor-based monitoring system: (**a**) Schematic of the photodiode-based monitoring system during the LPBF; (**b**) Placement of the photodiode sensor in the process chamber; and (**c**) The C10493-11 photodiode sensor.

**Figure 2 sensors-23-09793-f002:**
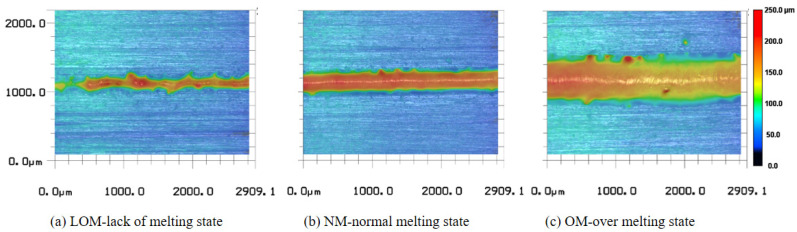
The three typical melting states obtained through laser confocal microscopy: (**a**) lack of melting state, (**b**) normal melting state and (**c**) over melting state.

**Figure 3 sensors-23-09793-f003:**
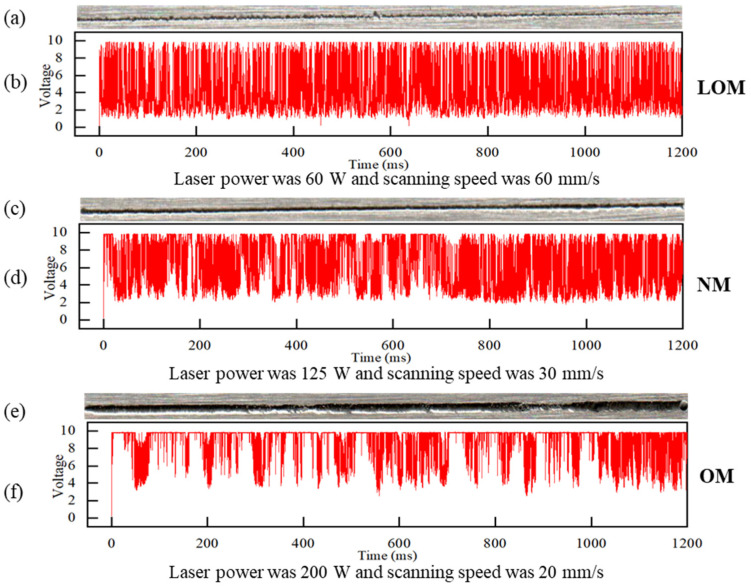
The top-view morphologies and the corresponding photodiode signals of three typical melting types of tracks. (LOM—lack of melting state, NM—normal melting state, and OM—over-melting states). (**a**) the optical appearance of the LOM, (**b**) the corresponding photodiode signal of LOM, (**c**) the optical appearance of the NM, (**d**) the corresponding photodiode signal of NM, (**e**) the optical appearance of the OM, (**f**) the corresponding photodiode signal of OM.

**Figure 4 sensors-23-09793-f004:**
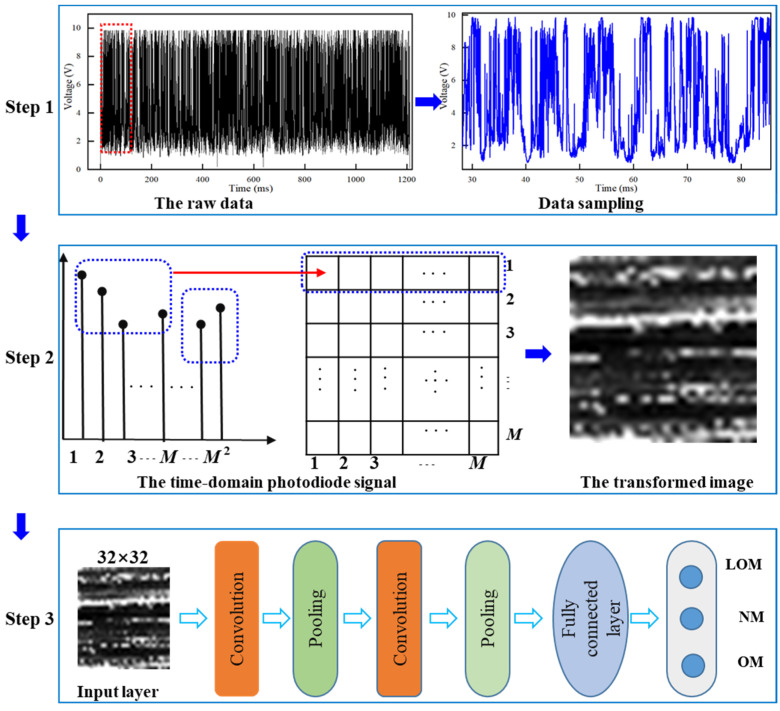
The procedure of the developed approach (the dots rectangles present the steps and the dashed rectangles presents the selected samples).

**Figure 5 sensors-23-09793-f005:**
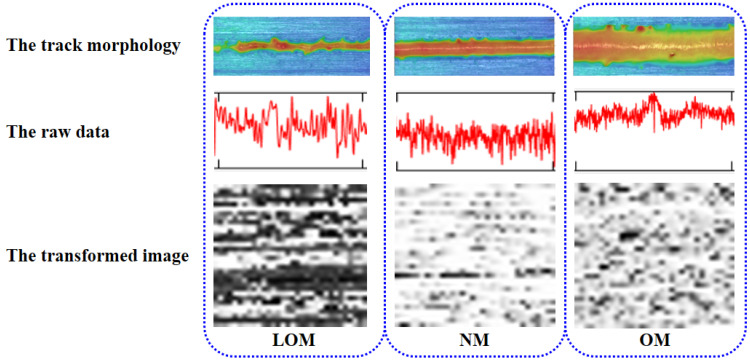
Three typical melting states and the converted images.

**Figure 6 sensors-23-09793-f006:**
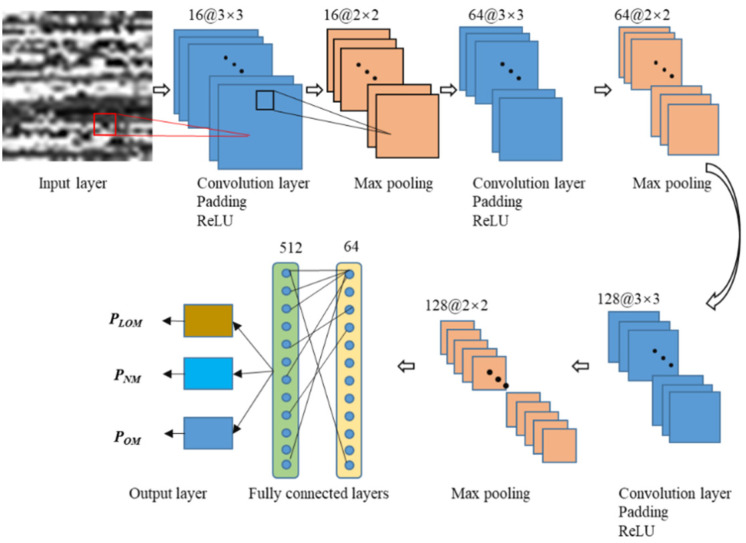
The structures of the proposed 2D CNN model ($P_{LOM}$ is the probability of lack of melting; $P_{NM}$ is the probability of normal melting; and $P_{OM}$ is the probability of over-melting).

**Figure 7 sensors-23-09793-f007:**
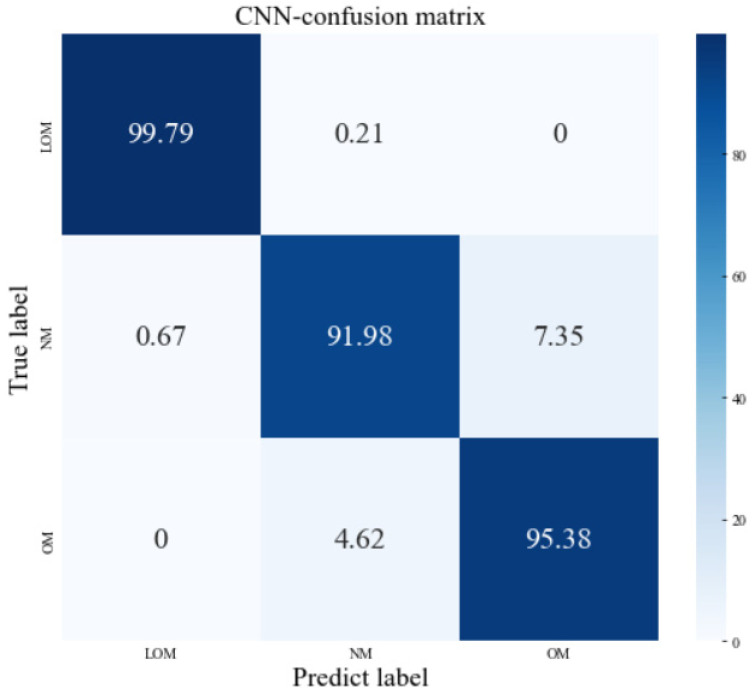
The confusion matrix of the test data set of original acoustic signal classification.

**Figure 8 sensors-23-09793-f008:**
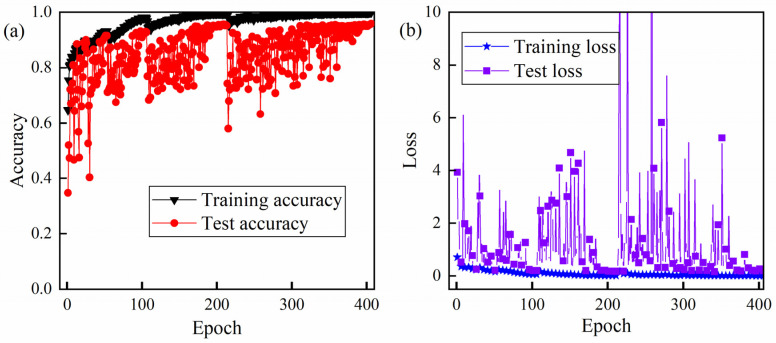
The curves of accuracy and loss of training and test: (**a**) the training and test accuracy and (**b**) the training and test loss.

**Figure 9 sensors-23-09793-f009:**
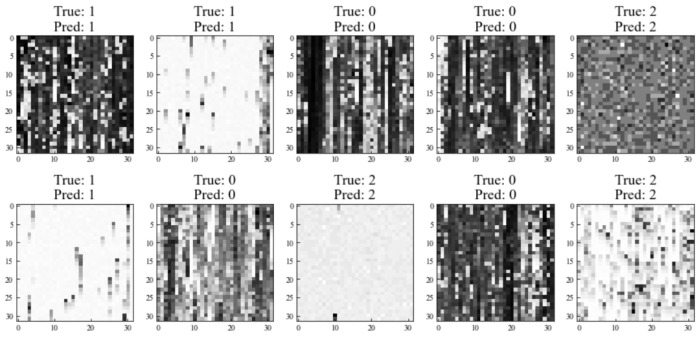
Visualization of proposed CNN testing results (‘true’ denotes the label, ‘pred’ denotes the predictive results. The number 0—lack of melting state; 1—normal melting state; and 2—over-melting states).

**Figure 10 sensors-23-09793-f010:**
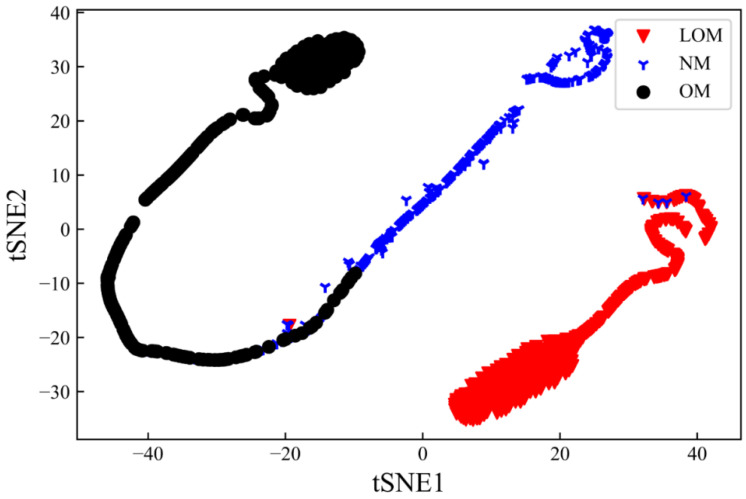
Results of the outputs of the second fully connected layer data after t-SNE computation.

**Figure 11 sensors-23-09793-f011:**
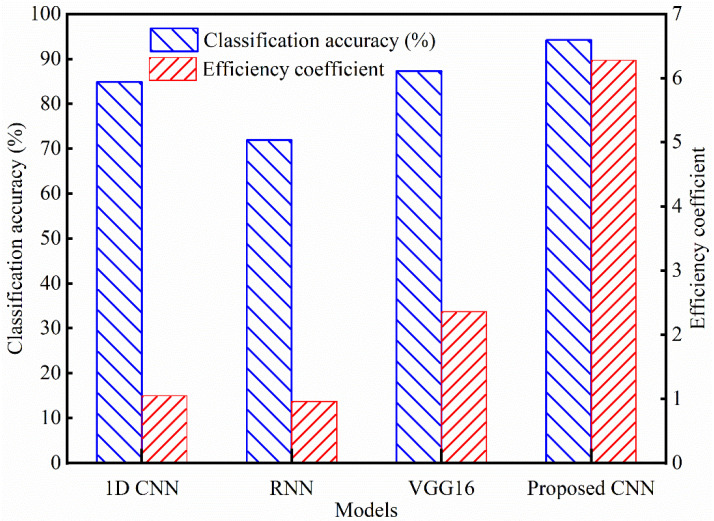
Classification accuracy (left y-axis) and efficiency coefficient (right y-axis) of the same hardware conditions.

**Table 1 sensors-23-09793-t001:** Nominal parameters of the LPBF machine.

Items	Values
Maximum print size	120 mm × 120 mm × 120 mm
Laser type	Fiber laser (RFL-C300L)
Heating bed temperature	473.15 K
Rated power	5 kW
Inert gas velocity	0.5–1.5 L/min
Spreading powder way	One-way scraper

**Table 2 sensors-23-09793-t002:** The elemental compositions of the 316L stainless steel powder.

Element.	C	Ni	Mn	S	P	Cr	Cu	Mo	Fe
Percent	<0.03	12.5–13	<2.00	<0.01	<0.02	17.5–18	<0.50	2.25–2.5	Balanced

**Table 3 sensors-23-09793-t003:** The LPBF processing parameters.

No.	Laser Power (W)	Scanning Speed (mm/s)	Spot Radius (um)	Volumetric Energy Density (J/mm^3^)	Melting States
1	50	50	80	49.8	LOM
2	60	60	80		
3	70	70	80		
4	120	28.8	80	207.3	NM
5	125	30	80		
6	130	31.2	80		
7	180	18	80	497.6	OM
8	200	20	80		
9	220	22	80		

Melting states are classified through the top-view morphology of the scan track. LOM—lack of melting; NM—normal melting; and OM—over-melting.

**Table 4 sensors-23-09793-t004:** Parameter configuration of the proposed CNN structure.

Layer	Type	Output	Number of Parameters
Input	Image data	32 × 32 × 1	0
Conv1	Convolution	32 × 32 × 16	320
Pool1	Max pooling	16 × 16 × 32	0
Conv2	Convolution	16 × 16 × 64	18,496
Pool2	Max pooling	8 × 8 × 64	0
Conv3	Convolution	8 × 8 × 128	73,856
Pool3	Max pooling	4 × 4 × 128	0
FC1	Fully Connected	64	131,136
FC2	Fully Connected	512	33,280
Output	Fully Connected	3	1539

**Table 5 sensors-23-09793-t005:** Comparison of different deep learning models.

Model	Classification Accuracy (%)	Computation Time (ms)	Efficiency Coefficient
1D CNN	84.92	81	1.05
RNN	71.92	75	0.96
VGG16	87.30	37	2.36
Proposed CNN	95.81	15	6.39

## Data Availability

Reader can contact the first or corresponding authors for data.

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
