# Peer review of "Monitoring of Single-Track Melting States Based on Photodiode Signal during Laser Powder Bed Fusion"

_sensors, 2023, doi:10.3390/s23249793_

Round 1

Reviewer 1 Report

Comments and Suggestions for Authors

The work titled "Monitoring of single track melting states based on photodiode signal during the laser powder bed fusion" is definitely a work of interest in LPBF. However, certain comments and questions about the Setup, Analysis methods, Experiments need to be addressed to clarify the novelty and the importance of the work. Some of my comments are given below:

1.    The literature cited seems slightly dated for the field, there have been a lot of recent advancements in LPBF from 2021-2023. I suggest authors review and refer some of the latest literature in the field.

2.    Why was a Photodiode sensor chosen as a Sensor of choice? When many recent works have shown the utility of using an off-axis camera, which can provide the image of the scan tracks directly – without the need of signal-to-image conversion. If the sampling rate of the Photodiode sensor was of primary importance, can you comment on its significance in the monitoring process?

3.    This work is focusing on processing the data offline – can the developed methods be implemented online? Meaning real-time? If the authors can show the utility of the developed methods and system in real-time then this would be of great use to the community and the field. Given that the CNN processing time for each sample is 15 ms, with the knowledge of signal-to-image conversion or an offline camera with direct image output would help develop online methods.

4.    In equation 1, why was the powder layer thickness not considered in the energy density calculations. Typically, energy density is categorized into Volumetric Energy Density (VED) or Linear Energy Density (LED). Just curious as to why this formula was adopted when the latest literature studies consider the layer thickness, hatch spacing (for multi-track printing), velocity and the laser power for calculating VED.

5.    Can you comment on the powder spreading accuracy? How certain are you that the powder is flatly spread on the Build platform – were any steps taken to measure the flatness of the build platform? The reason I ask this question is that this directly effects the balling and over melting due to improper laser heating. Of course, we can consider these as process anomalies, but need to be commented.

6.    In line 176, page 5, the authors mention results show good agreement with Zhang [26] what agreement are you referring to here? Could comment on the results from Zhang [26] as well?

7.    The photodiode signals in Figure 5 need to be explained well in the text, currently the explanation is very weak.

8.    Please elaborate the photodiode signal sampling mention in page 7, line198.

9.    Please move Equation 2 below the text appropriately – currently it appears in between the text.

10. Based on the Layer outputs provided in Table 4, the pooling layer should have a Stride of 2, to maintain the given outputs – please check this.

11. Further, the CNN architecture given in Fig 6 needs to be improved. Please check the text, the text says the number of channels/kernels in the Conv layers 32, 64, 128 but the Table 4 and Fig 6 say 16, 64, 128 – please clarify.

12. Regarding the data size and the training – your data size is 36 and the training set is 80%, which is approximately 29 samples, and the test samples are only 7! With such small data set size how can we establish the validity of the developed model for such a small test set? Could you provide more details about the model generalizability and modularity for other similar datasets.

13. Further, is it possible to extend the current dataset size?

14. Apart from the employed Neural Network model, were any other classification models tried? Such as SVM, or tree-based models? For such small datasets, these classification models could be tested as well and the accuracy of them can also be reported.

15. The conclusions and future works section can be expanded further, currently the conclusions are not very strong/convincing, and the future work is a bit vague.

16. The formatting in Section 3.3 needs to be improved, the variables in the text are out of place.

17. Line 230, Page 7, should be convert not invert. Please check other spelling mistakes as well, Fig.8 legend labels should be Training Losee not Taining and so on.

18. Fig 4 label is missing.

Comments on the Quality of English Language

The language can be improved - there are multiple spelling mistakes, please check them. Make sure all the Figure captions and the Table captions appropriately written and explained well.

Author Response

We thank the reviewer for the comments and suggestions, which were very valuable and very helpful for revising and improving our manuscript, as well as the important guiding significance to our research. We have carefully studied the concerns in the original manuscript. To adequately address this, we have made some changes. These changes are described in the item-by-item replies to the reviewer’s comments in the attached files. All the changes are annotated in the annotated version of the revised manuscript.

Reviewer 2 Report

Comments and Suggestions for Authors

In this manuscript, a simple and low-cost off-axial photodiode signal monitoring system was established to monitor the scanning tracks, and a CNN model was proposed to classify the melting state with the input of the grayscale images converted from the photodiode signal. Compared with some classical deep learning models, the proposed CNN could distinguish the melting states with higher classification accuracy and efficiency. The subject matter is of significant interest to scholars engaged in this particular area of study. However, some points need to be clarified. Therefore, the reviewer suggests major revision before this manuscript may be considered for publication. 

1.  The defects analysis in the LPBF process lacks clarity in the introduction. It is recommended to refer to this review “A review of the multi-dimensional application of machine learning to improve the integrated intelligence of laser powder bed fusion, Journal of Materials Processing Technology, 2023, 318, 118032”. Defects in the LPBF process are categorized based on their formation stage. During the powder laying process, defects mainly manifest as irregularities in the powder layer. On the other hand, defects formed during printing are more diverse and include spatter, balling, porosity, poor surface quality, cracks, and geometric distortion.

2: The focus of this manuscript is solely on the morphology analysis of the molten pool, but there is still a noticeable gap in the effectively control of defects during the LBPF process. Accordingly, how does the author consider the detection of actual defects (porosity, cracking, etc.) in the future? Or what is the reference value of this paper for the detection of actual defects during the LPBF process?

3: Although 1254744 data points were applied to generate 7200 grayscale images in this study, it should be noted that original data from only 36 experimental groups was relatively limited for training a CNN model effectively.

Table 3 presents energy density values, but the r value is not given in the table. It is suggested to add an additional column for r value.

5: Section 4.1 employs K-fold cross-validation for error estimation. However, no specific K value was mentioned or provided within this manuscript, so a specific explanation needs to be given.

6: The manuscript states that the number of channels in the first convolution layer is 32. However, Table 4 shows that the output of conv1 is 32×32×16. If we select 32 as the number of channels, it should be modified to 32×32×32, and similarly, 16@3*3 and 16@2*2 should be changed to 32@3*3 and 32@2*2 in Figure 6. Additionally, the fully connected layers are not properly depicted in Figure 6, and necessary modifications are recommended. Furthermore, Line 379 is incorrectly formatted.

Comments on the Quality of English Language

Minor editing of English language is required

Author Response

We thank the reviewer for the comments and suggestions, which were very valuable and very helpful for revising and improving our manuscript, as well as the important guiding significance to our research. We have carefully studied the concerns in the original manuscript. To adequately address this, we have made some changes. These changes are described in the item-by-item replies to the reviewer’s comments in the attached file. All the changes are annotated in the annotated version of the revised manuscript.

Round 2

Reviewer 1 Report

Comments and Suggestions for Authors

Thank you for answering all the review questions and improving the paper.

1. I just wanted to point out that the explanation for my Q4 in my first review is not satisfactory - I suggest changing the Energy density to Volumetric Energy Density for consistency with the field. Thanks for adding the additional citation for this.

2. Further, can you also add this explanation given in the responses to the text to make it more clear? You can shorten this if needed - "The collected signals from all experiments are scanned by running windows, of which the 1D data is transformed into 2D gray scale images that are used as inputs of the CNN classifier. To make the data of different categories balance. 34854 successive data points of each track experiment were applied to convert into 2D grayscale images, and then a total of 1254744 data points were augmented from 36 groups of experiments. Therefore, 200 images were obtained for each track and a total of 7200 image s were obtained for three categories states . 80% of the converted images in each class are stochastically selected for training (5760) and the remaining 20% for testing (1440). In this way, there are sufficient training samples to build and validate the state classification models."

3. Considering changing the title of 3.2 from "3.2. Signal-to-image method" to "Signal-to-image transformation methodology"

4. There are still some formatting errors in the document - please check page 9 and rectify them (point 16 from my previous review)

Comments on the Quality of English Language

English has been improved from the previous version - please check to see that all the headings and the figure captions are complete and descriptive.

Reviewer 2 Report

Comments and Suggestions for Authors

There are three main points of interest in this manuscript. First, a simple and low-cost off-axial photodiode signal monitoring system was established to monitor the scanning tracks,then the photodiode signal was converted to grayscale images. Second, a CNN model was proposed to classify the melting state with the input of the grayscale images converted from the photodiode signal. Lastly, compared with some classical deep learning models, the proposed CNN could distinguish the melting states with higher classification accuracy and efficiency. The subject matter is of significant interest to scholars engaged in this particular area of study. The manuscript has made satisfactory revisions to the previous comments. The reviewer recommends acceptance of this manuscript.

Comments on the Quality of English Language

Minor editing of English language required

Author Response

Thank you very much for your comments and suggestions. We appreciate for your warm work earnestly.